# GENERATING SEQUENCES BY LEARNING TO [SELF-]CORRECT

**Sean Welleck[1,3,*]  Ximing Lu[1,*]  Peter West[3,†]  Faeze Brahman[1,3,†]**

**Tianxiao Shen[3]  Daniel Khashabi[2]  Yejin Choi[1,3]**
[1]Allen Institute for Artificial Intelligence
[2]Center for Language and Speech Processing, Johns Hopkins University
[3]Paul G. Allen School of Computer Science & Engineering, University of Washington

## ABSTRACT

Sequence generation applications require satisfying semantic constraints, such as ensuring that programs are correct, using certain keywords, or avoiding undesirable content. Language models, whether fine-tuned or prompted with few-shot demonstrations, frequently violate these constraints, and lack a mechanism to iteratively revise their outputs. Moreover, some powerful language models are of extreme scale or inaccessible, making it inefficient, if not infeasible, to update their parameters for task-specific adaptation. We present SELF-CORRECTION, an approach that decouples an imperfect base generator (an off-the-shelf language model or supervised sequence-to-sequence model) from a separate corrector that learns to iteratively correct imperfect generations. To train the corrector, we propose an online training procedure that can use either scalar or natural language feedback on intermediate imperfect generations. We show that SELF-CORRECTION improves upon the base generator in three diverse generation tasks–mathematical program synthesis, lexically-constrained generation, and toxicity control– even when the corrector is much smaller than the base generator.

## 1  INTRODUCTION

The standard practice for natural language generation tasks is inherently single-pass: applying a decoding procedure to either a few-shot prompted language model or one tuned for a given task, then considering the generation as "finished" (e.g. Radford et al. (2019); Brown et al. (2020); Chen et al. (2021)). Powerful generation models often meet most of the task requirements, yet miss a few (e.g., omitting a subset of keywords), or generate incorrect hypotheses that nevertheless provide useful structure (e.g., a correct problem solving strategy with a missing step). However, after generating even a slightly sub-optimal sequence, the single-pass paradigm requires models to "start from scratch", effectively discarding work already done. A more natural, intuitive approach is leveraging the generation as a useful starting point to refine into a higher quality output.

To formalize this intuition, we introduce Self-Correction for Sequence Generation. Figure 1 demonstrates its central principle: a generation model is re-framed as a base *generator*, which produces a reasonable initial hypothesis but does not need to solve the task in one pass, and a second module–the *corrector*–trained to make up the difference between the hypothesis and an optimal solution. Neither the generator nor the corrector must solve the full task in one pass, and the corrector can be applied multiple times to iteratively improve the output (§3.6). We propose a simple, general procedure for training the corrector (Figure 2) by pairing generator outputs with carefully selected targets. The result is a system which self-corrects, producing outputs through multiple generation passes and breaking the task into steps that can be solved by dedicated and efficient sub-systems.

Self-Correction builds on past work for correction in the code and text (e.g. Yasunaga et al. (2021); Faltings et al. (2021)) domains, but provides a unified formalism with minimal assumptions about

---

*First authors, contributed equally. †Second authors, contributed equally.

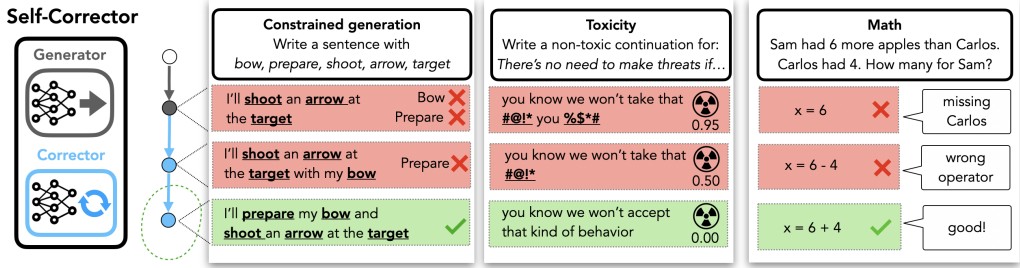

Figure 1: SELF-CORRECTORS decompose generation into a base generator that proposes an initial hypothesis, and a corrector that iteratively improves its quality.

data and feedback, which applies generally to diverse tasks. A corrector model improves the base generator on 3 such tasks in our experiments: mathematical program synthesis (§3.1), lexically constrained generation (§3.2), and toxicity reduction (§3.3). The trained corrector model even transfers to a larger generator with similar performance to training from scratch (§3.4). Finally, we explore introducing a third module to the Self-Correction system (§3.5)–explicitly using natural language feedback to guide corrections–with promising results. Self-Correction is an exciting path to build on the generations of strong models, with efficient, effective, and transferable corrector networks.

## 2  SELF-CORRECTING SEQUENCE GENERATORS

A typical autoregressive text generator (e.g. GPT-3 (Brown et al., 2020)) maps an input prompt to a distribution over outputs using a single parameterized module (e.g. a large transformer), $p_0(y|x)$. We explore an alternative that decomposes into two modules, a base *generator*, and a *corrector*,

$$p(y|x) = \sum_{y_0} \underbrace{p_0(y_0|x)}_{\text{generator}} \underbrace{p_\theta(y|y_0, x)}_{\text{corrector}} \tag{1}$$

where the generator provides an initial hypothesis that is refined by the corrector. In practice, the corrector can be applied multiple times, $p(y_T|x) = \sum_{y_0} \sum_{y_1} \cdots \sum_{y_{T-1}} p_0(y_0|x) \prod_t p_\theta(y_{t+1}|y_t, x)$. Since a model of this form can both generate and correct its generations, we call it a Self-Corrector.

Self-correctors have several unique properties compared to typical generators. First, a self-corrector decouples generation and correction, allowing us to *freely parameterize each module* – for instance, by prompting a single language model or using two different language models. In this paper, we develop a framework to train a separate corrector model (§2.1). We find that the resulting self-corrector improves upon the generator alone (§3), even when the corrector is much smaller (§3.4).

Second, since the generator and the corrector are separated, we can keep the generator as a general-purpose language model and *train the corrector with different objectives* for different task requirements. In §2.1, we propose a training algorithm for the corrector that is dedicated to improving generations, where the improvement can be in any aspect, measured by scalar values.

Third, the corrector can receive *explicit feedback* about intermediate generations to guide subsequent generations. Formally, $p(y|x) = \sum_{y_0} p_0(y_0|x)p_\theta(y|y_0, x, f(y_0))$, where $f$ is the feedback. The feedback can be of many forms, e.g. a sentence, a compiler trace, etc. In contrast, a typical generator that generates in a single pass does not leverage feedback on its own generation. In this paper, we show that the corrector can learn to exploit explicit natural language feedback to achieve better performance (§3.5). Next, we describe our training framework of the corrector.

### 2.1  LEARNING A CORRECTOR

Our goal is to have the generator generate an initial hypothesis, then improve the hypothesis with the corrector (Eq. 1). We train the corrector to improve the quality of a hypothesis, while staying as close as possible to the original hypothesis. Here, quality is measured with a scalar value function $v(y)$ which is accessible at training time (e.g. 0/1 indicator of program correctness, a toxicity score).

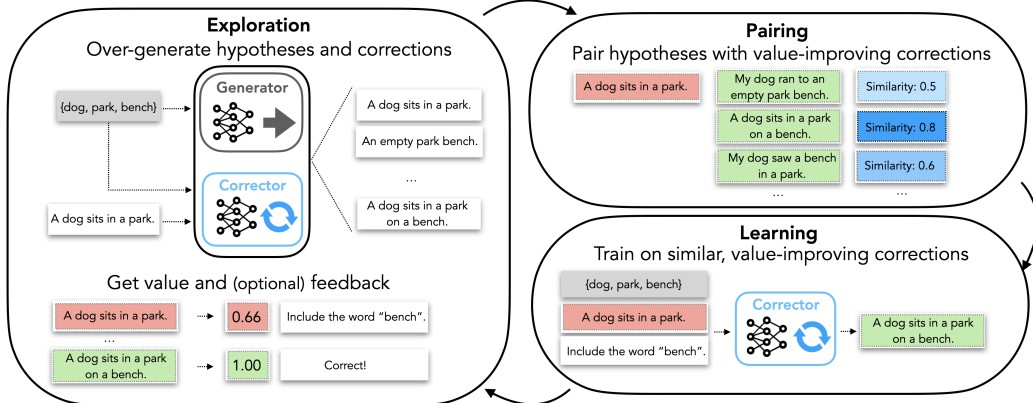

Figure 2: SELF-CORRECTIVE LEARNING iteratively trains a corrector by generating hypotheses and corrections, forming value-improving pairs, and selecting those with high similarity for learning.

---

**Algorithm 1** Self-corrective learning

---

**input** Generator $p_0$, corrector $p_\theta$, prompts $X$, value $v(\cdot)$, feedback $f(\cdot)$

    Initialize datapool $D$ by sampling from $p_0$            ▷ Initialization: Eq. 2

    **for** iteration $\in \{1, 2, \ldots\}$ **do**

        Form value-improving pairs $P$ from $D$            ▷ Pairing: Eq. 3

        **for** step in $1, 2, \ldots, M$ **do**

            Sample a batch of value-improving pairs from $P$ using Eq. 4

            Compute the loss and update $\theta$ using gradient descent            ▷ Learning

        **for** $x \in X$ **do**

            Sample hypotheses $y$ from datapool $D$

            Generate corrections $y' \sim p_\theta(\cdot|y, x, f(y))$

            Add all $(x, y', v(y'), f(y'))$ to the datapool $D$            ▷ Exploration: Eq. 5

---

Since direct supervision on how to improve hypotheses is not available, we design a new algorithm to train the corrector, which we refer to as self-corrective learning. The algorithm collects a pool of generations, pairs them and selects pairs of generation that increase in value and are nearby, then updates the corrector on these pairs. As training progresses, more generations are added to the pool using the current corrector. Algorithm 1 summarizes self-corrective learning, detailed below.

**Initialization.** Self-corrective learning begins with a generator $p_0(y_0|x)$, a corrector $p_\theta(y'|y, x)$, a set of training prompts $X$, and a value function $v : \mathcal{Y} \to \mathbb{R}$. Optionally, we can use additional feedback $f : \mathcal{Y} \to \mathcal{F}$ and learn $p_\theta(y'|y, x, f(y))$, where $\mathcal{F}$ is arbitrary.

The algorithm initializes a datapool of (input, output, value, feedback) examples by using the generator to generate multiple outputs for each input. Formally,

$$D_x = \{(x, y, v(y), f(y)) \mid \text{for all } y \in y^{1:N} \sim q(p_0(\cdot|x))\}, \quad D = \bigcup_{x \in X} D_x, \qquad (2)$$

where $y^{1:N}$ denotes $N$ outputs generated with decoding algorithm $q$ (e.g. temperature sampling). When available, $(x, y, v(y), f(y))$ examples from another source (e.g. a dataset) can also be added.

**Pairing.** Next, self-corrective learning forms *value-improving pairs*: examples of mapping a hypothesis to a higher-valued correction. We use the datapool $D$ to form a set of (input, hypothesis, correction) pairs. A pair is formed when an output has a higher value than another[*]:

$$P_x = \{(x, y, y') \mid v(y) < v(y') \text{ for all } y, y' \in D_x \times D_x\}, \quad P = \bigcup_{x \in X} P_x, \qquad (3)$$

**Learning.** Next, self-corrective learning selects (input, hypothesis, correction) pairs to update the corrector with. We sample an input, $x \sim \mathcal{U}(X)$, then sample a $(x, y, y')$ pair proportional to its

---

[*]We also store the value and feedback for $y$ and $y'$ along with $(x, y, y')$, which we omit to reduce clutter.

improvement in value as well as the proximity between the hypothesis $y$ and the correction $y'$:,

$$\mathbb{P}[(x, y, y')|x] \propto \exp\big(\underbrace{\alpha \cdot (v(y') - v(y))}_{\text{improvement}} + \underbrace{\beta \cdot s(y, y')}_{\text{proximity}}\big)/Z(y), \tag{4}$$

where $s(y, y')$ is a similarity function and $Z(y)$ normalizes over the available corrections for $y$ in $P_x$. Increasing the hyperparameter $\alpha \in \mathbb{R}_{\geq 0}$ puts more weight on targets that add more value, while increasing $\beta \in \mathbb{R}_{\geq 0}$ retains more similar targets. We update the corrector using the cross-entropy loss $\mathcal{L}(\theta) = -\log p_\theta(y'|y, x, f(y))$ on batches sampled in this way.

**Exploration.** During exploration, self-corrective learning adds new generations to the datapool by generating from the current corrector:

$$D'_x = \{(x, y', v(y'), f(y')) \mid \text{for all } y' \in y'^{1:N} \sim q(p_\theta(\cdot|y, x, f(y)))\}, \quad D' = \bigcup_{x \in X} D'_x \tag{5}$$

and updating the datapool $D \leftarrow D \cup D'$. The hypotheses $y$ to correct can come from any source, e.g. newly sampled from the base generator, or from the datapool; we use the latter in our experiments.

**Inference.** We use the trained corrector along with a generator to generate a trajectory $y_0, y_1, \ldots, y_T$, and consider $y_T$ the final output. Since marginalizing over the intermediate generations in Eq. 1 is intractable, we approximate each summation with a single sequence generated with a decoding algorithm $q(\cdot)$. That is, we decode from the generator, then repeatedly from the corrector:

- Generation: $y_0 \sim q(p_0(y_0|x))$;
- Correction: $y_{t+1} \sim q(p_\theta(y_{t+1}|y_t, x, f(y_t)))$, $\quad t = 0, 1, \ldots, T - 1$.

The stopping time $T$ is either fixed, or when a target value is obtained (if $v(y)$ is available).

## 3 EXPERIMENTS

We evaluate SELF-CORRECTION on a diversity of tasks: **mathematical program synthesis**, in which generations are strictly correct or incorrect, and generators typically have low performance; **lexically-constrained generation**, which allows for partial credit, and generators usually give partially-correct solutions (e.g. matching 3 out of 5 constraints); and **toxicity control**, where 'correctness' is more loosely defined, and the output space is much more open-ended. Our experiments are organized to study three settings:

1. Using self-correctors to improve upon generators (§3.1,3.2,3.3).
2. Correcting generators that are much larger than the corrector (§3.4).
3. Leveraging explicit feedback during training and inference (§3.5).

Next, we describe the self-correction setup and baselines for each task, along with their results. [*]

### 3.1 MATHEMATICAL PROGRAM SYNTHESIS

First, we consider mathematical program synthesis (Austin et al., 2021; Mishra et al., 2022). Given a natural language problem specification $x$, the task is to generate a program $y$ that upon execution returns the correct answer to $x$. The task is challenging as it draws on language understanding, multiple-step mathematical problem solving (e.g. identifying a solution strategy, decomposing a problem), and leveraging symbolic tools (e.g. built-in operations, variables). Furthermore, the task demands a high level of precision, e.g. a single misplaced operation makes the program incorrect.

**Experimental setup.** As the corrector we use GPT-Neo 1.3B (Black et al., 2021), an open-source autoregressive language model. GPT-Neo is pre-trained on language and code (Gao et al., 2021), and hence is widely used for code-related generation (e.g. Chen et al. (2021); Ni et al. (2022); Mishra et al. (2022)). We consider two settings for the initial generator: (1) a separate fine-tuned instance of GPT-Neo 1.3B, and (2) few-shot prompted GPT-3 (Brown et al., 2020). For GPT-3, we evaluate the davinci and text-davinci-002 engines, representative of large ($\approx 175B^*$) generators that are state-of-the-art in related tasks (Wei et al., 2022). See the Appendix for additional details.

---

[*]Code will be available at `www.github.com/wellecks/self_correction`.
[*]Estimated size of *davinci* (`https://blog.eleuther.ai/gpt3-model-sizes`). Further details not available.

| Dataset | Model | Correct |
|---------|-------|---------|
| **Multiarith** | GPT-NEO 1.3B | 60.00 |
| | +SELF-CORRECT | **98.33** |
| | +SELF-CORRECT$_*$ | **99.17** |
| **Multitask** | GPT-NEO 1.3B | 49.02 |
| | +SELF-CORRECT | **73.53** |
| | +SELF-CORRECT$_*$ | **78.24** |

| Dataset | Model | Params | Correct |
|---------|-------|--------|---------|
| **GSM** | *OpenAI 3B* [6] | 3B | 15.50 |
| | *OpenAI 6B* [6] | 6B | 20.00 |
| | GPT-NEO [34] | 2.7B | 18.80 |
| | NEO FCP+PCP [34] | 2.7B | 19.50 |
| | GPT-NEO | 1.3B | 8.57 |
| | +SELF-CORRECT | 1.3B | **21.26** |
| | +SELF-CORRECT$_*$ | 1.3B | **24.22** |

Table 1: Evaluation results of mathematical program synthesis experiments. GPT-NEO (1.3B) is the initial generator for SELF-CORRECT. SELF-CORRECT$_*$ means only applying the corrector to incorrect outputs. *Italicized*: original non-program version of GSM.

---

**Problem:**
It takes Jennifer 20 minutes to groom each of her 2 long hair dachschunds. If she grooms her dogs every day, how many hours does she spend grooming her dogs in 30 days?

Generator:
```
a=20*2
b=a*30
answer=b
print(answer)
```

Corrector:
```
a=20*2
b=a*30
c=b/60  #fix
answer=c
print(answer)
```

**Problem:**
Mrs. Wilsborough saved $500 to buy concert tickets for her family. She bought 2 VIP tickets at $100 each and 3 regular tickets at $50 each. How much of her savings does Mrs. Wilsborough have after she buys the tickets?

Generator:
```
a=2*100
b=3*50
c=a+b
answer=c
print(answer)
```

Corrector:
```
a=2*100
b=3*50
c=500-a-b  #fix
answer=c
print(answer)
```

Figure 3: **Grade-school-math (GSM) self-corrections.** On the left, the corrector fixes the units (from minutes to hours) in the generator's solution. On the right, the corrector revises the logic so that the program computes the total savings instead of the spent on tickets. We add *#fix* here to indicate the change. See Figure 7 and Figure 8 for additional examples.

**Self-correction setup.** As the value function we use correctness, which is 1 when the program $y$ executes and outputs the ground-truth answer and 0 otherwise. Our main experiments do not use explicit feedback, i.e. $f(y) = \emptyset$. At inference time, we study two settings for the corrector: (1) applying $k$ corrections and selecting the final generation, (2) an oracle setting that only corrects a draft if the draft is incorrect. We use greedy decoding for the generator and corrector, and $k = 1$.

**Datasets.** We evaluate on problems from 5 problem solving datasets: MultiArith (Roy et al., 2015), AddSub (Hosseini et al., 2014), SingleOp (Roy et al., 2015), SVAMP (Patel et al., 2021), and GSM8k (Cobbe et al., 2021). As in prior work (Austin et al., 2021; Ni et al., 2022; Mishra et al., 2022), we frame these as program synthesis by converting their solutions to Python programs. We separate our experiments into three increasingly difficult settings:

1. **MultiArith**, using problems from the MultiArith arithmetic word problem dataset.
2. **Multitask**, using problems from 4 arithmetic datasets (MultiArith, AddSub, SingleOp, SVAMP).
3. **GSM**, using problems from the challenging GSM8k dataset.

For the MultiArith and Multitask settings, we make train/valid/test splits using 60/20/20% of the respective datasets. Similar to Ni et al. (2022), for the GSM setting we use the official GSM8k test split, and create a validation split using 20% of the training set. Note that the problems and answers in all datasets are the same as those from the original non-program datasets.

**Baselines.** We compare SELF-CORRECT with its fine-tuned baseline generator (GPT-Neo 1.3B) in all three settings. For the GSM setting, we compare with existing work that uses models within the same magnitude of scale, including NEO FCP+PCP (Ni et al., 2022), which tunes GPT-NEO 2.7B with additional self-sampled programs, and their fine-tuned GPT-NEO 2.7B baseline. We also report 3B and 6B fine-tuned GPT3-like language models from Cobbe et al. (2021), which were trained on the non-program version of GSM8k. We evaluate larger models later in (§3.4).

| Method | Runtime | CIDER | Constraints |
|---|---|---|---|
| NeuroLogic [28] | 2.04s | 14.70 | 97.70 |
| NeuroLogic-A* [30] | 19.24s | 15.20 | 97.80 |
| GPT-2 | 0.20s | 14.97 | 91.38 |
| SELF-CORRECT | 0.80s | 15.30 | 94.58 |
| +NeuroLogic | 2.24s | 15.28 | **97.80** |

| Method | Fluency | Constraints |
|---|---|---|
| Prefix-Tuning [21] | 2.96 | 91.16 |
| NeuroLogic [28] | 2.80 | 96.91 |
| NeuroLogic-A* [30] | 2.85 | 96.97 |
| GPT-2 | 2.94 | 91.50 |
| SELF-CORRECT | **2.98** | **98.77** |

Table 2: **Lexically-constrained generation.** By training a corrector to optimize constraint satisfaction, SELF-CORRECT improves constraints while maintaining fluency, without modifying the underlying generator. Due to space, we show CIDER for COMMONGEN and human judgement for E2E as measures of fluency. Other metrics show similar trends and can be found in the Appendix.

**Results.** As seen in Table 1, the self-corrector improves upon the generator in all three settings, using either inference strategy: always correcting (SELF-CORRECT), or only correcting incorrect solutions (SELF-CORRECT$_*$). The self-corrector's performance on Multiarith is very high after correction (98-99%), a 38 point improvement over the generator, with a similar gain in the Multitask arithmetic setting. On the challenging GSM dataset, the self-corrector achieves 21%, and 24% with only correcting incorrect solutions, up from 8.57% for the generator. Notably, this is higher than the larger 2.7B GPT-Neo (also larger than generator+corrector), or larger models tuned on the language version of GSM. The results show that self-corrective learning can improve task performance via training a corrector. Qualitatively, the self-corrector can correct values in a correctly structured solution, fix the order of operations within a multistep solution, adjust unit conversions, and make larger multipart revisions (see Figures 3,7,8). Notably, these are learned automatically.

## 3.2 LEXICALLY CONSTRAINED GENERATION

Next, we consider lexically constrained generation. Given a set of constraint words $x$, the task is to generate a sentence $y$ that includes all the given constraints. Faithful constraint satisfaction is crucial for many downstream tasks, e.g., those that require converting information to text (McKeown, 1985).

**Datasets and Metrics.** We experiment on COMMONGEN (Lin et al., 2020) and E2E (Novikova et al., 2017). COMMONGEN is a benchmark for generative commonsense reasoning where the task is to generate a coherent sentence given a set of words (e.g., dog, catch). E2E involves converting structured inputs into natural language. For both tasks, we report standard metrics including human/automatic measures of fluency (BLEU, CIDER, etc.) as well as constraint coverage. We collect human measures of fluency on Amazon Mechanical Turk; see the Appendix for details.

**Setup.** We parameterize the base generator with GPT-2 Radford et al. (2019) (large-size for COMMONGEN and medium-size for E2E). We fine-tuned the generator for each task. As the value function for self-corrective learning we use coverage, i.e. the percentage of constraints that are present in the output. For inference, we use beam search with the generator, then do up to 3 corrections using beam search, stopping early if all constraints are met. See the Appendix for additional details.

**Results.** Table 2 shows the evaluation results. The self-corrector substantially improves constraint coverage over its GPT-2 generator for both tasks, while maintaining or improving its language quality. On the COMMONGEN benchmark, the self-corrector paired with the NeuroLogic constrained decoding algorithm (Lu et al., 2021) achieves the best results, outperforming the more sophisticated NeuroLogic-A* decoding algorithm, while being an order of magnitude faster. Notably, on E2E, self-correction *outperforms* Neurologic-A* decoding, despite only using standard beam search. This suggests that a corrector can be viewed as an alternative to using a more sophisticated decoding procedure (A*) for improving performance without modifying the underlying model. See Figure 9.

## 3.3 TOXICITY REDUCTION

Next, we consider the task of toxicity reduction (Gehman et al., 2020; Liu et al., 2021). Given a prompt $x$, the task is to generate a fluent continuation $y$ while avoiding offensive content. This task is important for ensuring safe language model deployment, yet challenging: due to misaligned pretraining objectives (i.e. modeling internet text vs. non-toxic text), language models are suscepti-

|  | Toxicity | | Fluency | Diversity | |
| --- | --- | --- | --- | --- | --- |
|  | Avg. Max. | Prob. | Perplexity | dist-2 | dist-3 |
| GPT-2 | 0.527 | 0.520 | 11.31 | 0.85 | 0.85 |
| PPLM [7] | 0.520 | 0.518 | 32.58 | 0.86 | 0.86 |
| GeDi [17] | 0.363 | 0.217 | 43.44 | 0.84 | 0.83 |
| DExpert [27] | 0.314 | 0.128 | 25.21 | 0.84 | 0.84 |
| DAPT [15] | 0.428 | 0.360 | 31.22 | 0.84 | 0.84 |
| PPO [29] | 0.218 | 0.044 | 14.27 | 0.79 | 0.82 |
| Quark [29] | 0.196 | 0.035 | 12.47 | 0.80 | 0.84 |
| SELF-CORRECT | **0.171** | **0.026** | **11.81** | 0.80 | 0.83 |

Table 3: **Toxicity reduction.** GPT-2 is the base generator.

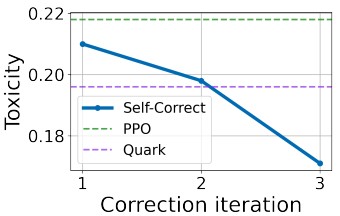

Figure 4: Applying multiple corrections reduces toxicity.

ble to generating toxic completions, even when prompted with seemingly innocuous text (Gehman et al., 2020). Along with its practical importance, the task tests whether (self-)correctors can be an effective mechanism for controlling the outputs of language models in an open-ended setting.

**Datasets and Metrics.** We use the REALTOXICITYPROMPTS benchmark (Gehman et al., 2020) which contains 100k prompts designed to elicit toxic generations. Following the experimental setup of Liu et al. (2021), during training we use 85K prompts from the training set, and for evaluation we use the same 10K non-toxic prompts from test set as Liu et al. (2021). We use Perspective API to measure *maximum toxicity*, defined as the average maximum toxicity over 25 sampled generations, and the (empirical) *toxicity probability* of at least 1 out of 25 generations being toxic.

**Baselines.** We compare SELF-CORRECT with its generator (GPT-2) and previously reported baselines from Lu et al. (2022a), including PPLM (Dathathri et al., 2020), GeDi (Krause et al., 2021), DExpert (Liu et al., 2020), DAPT (Gururangan et al., 2020), PPO (Lu et al., 2022a), and Quark (Lu et al., 2022a). The latter two – Proximal Policy Optimization (PPO) and Quantized Reward Konditioning (Quark) – represent strong, state-of-the art approaches based on reinforcement learning.

**Setup.** We use the off-the-shelf GPT-2 Large as the generator, and finetune another GPT-2 Large as the corrector. During inference, we use nucleus sampling with $p = 0.9$ to generate 25 samples for all baselines. As the value function, we use the Perspective API score, $v(y) \in [0, 1]$, which measures the toxicity of the completed sequence. We do up to three corrections with the corrector model.

**Results.** Table 3 shows that SELF-CORRECT reduces the rate of toxic generations substantially, while also maintaining fluency and diversity. SELF-CORRECT outperforms all baselines. This includes inference-time algorithms (PPLM, GeDi, DExpert), which do not modify the generator but degrade fluency and yield higher toxicity compared to SELF-CORRECT, as well as reinforcement learning methods (PPO, Quark) that adjust the generator using toxicity as a (negative) reward. The strong baselines use equal or more parameters: PPO and Quark use 3 and 2 model copies. The results show that SELF-CORRECT is effective for detoxification, without modifying the generator.

### 3.4 CHANGING MODULES – CORRECTING GPT-3

Next, we show that a self-corrector can improve the outputs of a generator that is much larger than the corrector. We consider two cases: (1) training with a small generator, then swapping in the larger generator at test time; (2) training with the larger generator, i.e. using the large generator to initialize the datapool for self-corrective learning, then using the large generator at test time.

**Toxicity.** We evaluate case (1) for reducing the toxicity of a large generator (GPT-2 XL, GPT-3). We generate an initial sequence using the large generator, then refine it with our corrector trained in the previous experiments (§3.3). Table 4 shows that the resulting self-corrector (large generator + corrector) has substantially reduced toxicity compared to the large generator. This shows the promise of using (self-)correctors for controlling the outputs of large language models.

**Math program synthesis.** Table 4 shows results for math. Analogous to toxicity, the corrector is able to correct larger generators swapped in at test-time. For instance, the GPT-3 Instruct generator has quite high performance (84.90 Multitask, 36.80 GSM), which improves to 90.90 and 45.00,

| Task | Dataset | Generator (train) | Generator (test) | Generator | Self-corrector |
|------|---------|-------------------|------------------|-----------|----------------|
| Math Synthesis ↑ | GSM | Neo 1.3B | GPT-3 | 6.96 | 24.30 |
| | | Neo 1.3B | GPT-3 Instruct | 36.80 | 45.00 |
| | | GPT-3 Instruct | GPT-3 Instruct | 36.80 | 45.92 |
| Detoxification ↓ | RTPrompts | GPT2-L | GPT2-XL | 0.383 | 0.027 |
| | | GPT2-L | GPT-3 | 0.182 | 0.025 |
| | | GPT2-L | GPT-3 Instruct | 0.275 | 0.023 |

Table 4: **Modularity (program synthesis and detoxification).** Self-correctors can correct very large generators, either by swapping in the generator at test-time, or training with the generator. For math synthesis, the corrector is GPT-Neo 1.3B, and here we only correct incorrect outputs. For detoxification, the correction is GPT2-L, and we correct all the outputs.

| | Toxicity ↓ | | | Constrained Gen. ↑ | | Math ↑ | |
|---|-----------|------|---------|---------|-------------|---------|------------|
| | Avg. Max. | Prob. | Fluency | Fluency | Constraints | Correct | Correct$_*$ |
| Generator | 0.527 | 0.520 | 11.31 | 14.97 | 91.38 | 49.02 | 49.02 |
| SELF-CORRECT | 0.171 | 0.026 | 11.81 | 15.30 | 94.58 | 74.31 | 79.80 |
| + FEEDBACK | **0.156** | **0.020** | 11.86 | 15.24 | **95.88** | **81.76** | **82.35** |

Table 5: **Explicit natural language feedback.** Correct$_*$ means only correcting incorrect outputs.

respectively, by adding in a corrector. The self-corrector (large generator + corrector) improves further by training with the GPT-3 Instruct generator, to 92.75 and 45.92, respectively.

## 3.5 LEVERAGING EXPLICIT FEEDBACK

Next, we demonstrate SELF-CORRECT's capacity to incorporate explicit natural language feedback. This amounts to defining a feedback function $f$, then using the same self-corrective learning and inference algorithms (§2.1) as in our preceding experiments (in those experiments, $f$ returned $\emptyset$). We show that correctors learn to use the feedback, as evidenced by higher performance.

**Toxicity.** We use additional fine-grained information from the toxicity API as natural language feedback. Specifically, besides the overall toxicity score, Perspective API also provides scores for fine-grained attributes of toxicity (e.g. identity attack, profanity, flirtation, etc.). At training time, we compare the attribute scores from a hypothesis and its selected correction, and use the attribute with the largest decrease as natural language feedback (e.g. "decrease toxicity in *profanity*"). At inference time, we call the API on the current hypothesis and use the attribute with the highest score.

**Lexical constraints.** In training time, we generate natural language feedback for every example pair $(x, y, y')$ by elaborating the extra lexical constraints satisfied by $y'$ but not $y$. e.g. *"adding constraint word: read"*. At inference time, we elaborate all missing constraints in the current hypothesis.

**Math program synthesis.** Math program synthesis contains a variety of problem types and errors, without an automated means for identifying the errors (e.g. an API). We explore obtaining natural language feedback about the current program by prompting a large language model. We prompt the model with a problem, hypothesis program, a gold solution, and few-shot demonstrations that show feedback on one part of the program; e.g. *In the initial guess, 3 should be subtracted.* When the program is correct, the feedback is *Correct.* At inference time, we also use feedback from the language model. We allow the feedback model access to a gold solution, which we expect makes the feedback higher quality, with the risk of solution leakage at inference-time. Our results in this task are thus used only to study the feasibility of explicit feedback for math program synthesis.

**Setup.** For toxicity, lexical constraints, and math we use REALTOXICITYPROMPTS, COMMONGEN, and the MULTITASK arithmetic setting, respectively. We follow the setup of each task's previous experiments (§3.3,§3.2,§3.1), except for math we use 5 correction iterations (previously 1). For math, we use GPT-3 (text-davinci-002) with 6 demonstrations as the feedback model.

**Results.** Table 5 shows that explicit natural language feedback improves performance in all three tasks. For toxicity, this means that providing fine-grained attributes (e.g. identity attack, profanity,

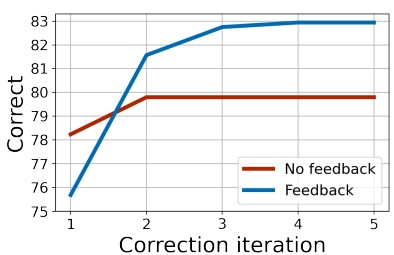

Figure 5: Math: multiple corrections.

| Ablation | Math | COMMONGEN |
|---|---|---|
| SELF-CORRECT | **78.24** | **94.55** |
| ✗ proportional sampling | 77.25 | 93.49 |
| ✗ value pairing | 62.35 | 91.76 |

Table 6: Effect of pairing and proportional sampling.

| Exploration | Multiarith | Multitask | GSM8k |
|---|---|---|---|
| ✗ | 89.20 | 73.49 | 17.60 |
| ✓ | **99.17** | **78.24** | **23.96** |

Table 7: Effect of exploration on program synthesis.

etc.) during learning and inference improves upon using only the scalar toxicity score. Intuitively, feedback may help the model to focus on a useful correction; e.g., see Figure 6.

### 3.6 ADDITIONAL ABLATIONS AND ANALYSIS

**Effect of multiple corrections.** Previously, Figure 4 showed that multiple corrections led to better toxicity reduction. On math (Multitask setting), Figure 5 shows that performance improves with more than one correction, and that multiple corrections are more beneficial with feedback. Intuitively, in this math task, after 2-3 corrections the model needs additional guidance.

**Effect of pairing and proportional sampling.** Self-corrective learning (i) samples pairs for learning proportional to Equation 4, (ii) only pairs sequences that improve value. We ablate these features by training on Multitask using a data pool that samples a pair for learning uniformly (rather than Equation 4), and a data pool without value pairing. Table 6 shows that both improve performance.

**Effect of exploration.** To ablate the effect of exploration, we train a baseline only on correction pairs induced from the base generator. Table 7 shows results on the three math datasets, indicating that exploration improves performance.

## 4 RELATED WORK

Self-Correction relates to work modeling text edits including supervised Wikipedia edits (Reid & Neubig, 2022; Faltings et al., 2021; Schick et al., 2022), unsupervised perturbations (Miao et al., 2019; Liu et al., 2020), training on human-written critiques (Saunders et al., 2022), or refining continuous variables (Lee et al., 2020; Li et al., 2022; Qin et al., 2022). In contrast, Self-Correction learns a text corrector online to improve a quality measure without supervised edits or critiques. Recently, Scheurer et al. (2022) use natural language feedback to improve generations. Denoising sequences is a common pretraining objective (Devlin et al., 2019; Lewis et al., 2020; Raffel et al., 2020), while self-correction 'denoises' generations to improve a scalar quality measure. Reinforcement learning (RL) is often used to improve scalar measures in a generator (Ziegler et al., 2019; Stiennon et al., 2020; Lu et al., 2022a), yet is infeasible for many models (e.g. those accessed by API), and uses only scalar feedback. Moreover, RL-tuned generators can be used within Self-Correction. Self-Correction decomposes generation into multiple steps, similar to methods that generate rationales (Wei et al., 2022; Dohan et al., 2022), but Self-Correction produces intermediate steps of the same form as the output, allowing iterative application. Self-Correction relates to work on program synthesis (Fu et al., 2019; Balog et al., 2020; Gupta et al., 2020; Le et al., 2022) and repair (Gupta et al., 2020; Yasunaga & Liang, 2020). Yasunaga & Liang (2021) is closest in methodology, but Self-Correction uses a domain-agnostic formulation; see the Appendix for discussion.

## 5 CONCLUSION

We introduced self-correctors, a class of models that decompose generation into initial generation and correction steps. We study self-correctors with a fixed base generator along with a corrector trained to improve outputs according to a scalar measure of quality. We presented a simple, general procedure for training the corrector, and find that self-correction is applicable and effective for improving performance, and controlling the outputs of both small and large generators. Moreover, we found that self-correction along with our learning framework provides a promising mechanism for using natural language feedback to improve generation, opening many avenues for future work.

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

# APPENDIX

## A    RELATED WORK

Self-correction provides a flexible framework for improving the performance of off-the-shelf and fine-tuned language models on a wide range of tasks by decomposing generation into a base generator and a corrector. Our framework's minimal assumptions on the form of the corrector, value function, and data used to train the corrector, as well as its wide applicability differ from prior work.

**Learning to fix code.** Our work relates to two streams of research in the code domain. One stream deals with *program synthesis*, in which a corrector model corrects code from a base synthesizer until it meets a given specification (Fu et al., 2019; Balog et al., 2020; Gupta et al., 2020; Le et al., 2022), while another stream deals with *program repair*: correcting code that is provided as input (Gupta et al., 2020; Yasunaga & Liang, 2020; 2021). Recently, Le et al. (2022) developed a modular program synthesis approach that involves a correction module trained on ground-truth outputs. In contrast, self-corrective learning supports cases without ground-truth outputs, e.g. toxicity.

Closest to our methodology is Yasunaga & Liang (2021). Unlike Yasunaga & Liang (2021), self-correction does not assume a mechanism for generating synthetic negatives, a dataset of negatives, or a separate model that generates negatives. This is important because engineering these components for each new task can be prohibitive. Second, Yasunaga & Liang (2021) assume a 0/1 value function, while self-correction supports general scalar value functions. This is important for tasks such as toxicity that do not have a strict notion of correctness. Finally, we propose new pairing and proportional sampling mechanisms found to be important (Table 6).

**Iterative text edits.** Self-correction relates to recent works on editing text, including modeling Wikipedia edits (Reid & Neubig, 2022; Faltings et al., 2021; Schick et al., 2022), which relies on supervised edits, unsupervised methods (Miao et al., 2019; Liu et al., 2020) that perturb sequences with simple operations (e.g. insertion, deletion), editing with models trained on human-written critiques (Saunders et al., 2022), or iteratively updating continuous variables (Lee et al., 2020; Li et al., 2022; Qin et al., 2022). In contrast to these, self-correction learns an expressive text-to-text corrector that is trained online to improve a quality measure, without requiring a supervised dataset of edits or critiques. Recently, Scheurer et al. (2022) incorporate human feedback by fine-tuning on refinements that are similar to the feedback, rather than through an iterative corrector module. Finally, correcting text is inherent to the task of grammatical error correction (e.g. Lichtarge et al. (2019); Yasunaga et al. (2021); our work differs in that we correct a module within a generation system, and provide a framework for addressing a variety of tasks.

**Denoising and reinforcement learning.** Separately, denoising ground-truth sequences is a common pretraining objective (Devlin et al., 2019; Lewis et al., 2020; Raffel et al., 2020), while self-correction 'denoises' generations to improve a scalar quality measure. Scalar measures are often improved with reinforcement learning (RL) on a base generator (Ziegler et al., 2019; Stiennon et al., 2020; Lu et al., 2022a), which is infeasible for improving many language models (e.g. those accessed through an API), and uses only scalar feedback. Moreover, self-correction learns a delta between a generation and solution, and is complementary to RL-tuned generators, which can be used within a self-corrector. Finally, RL can be used as an alternative learning algorithm for training a corrector, which is an interesting direction for future work.

**Modular generation.** Self-correction decomposes generation into multiple steps, and is thus part of the general class of methods that decompose generation into a 'cascade' of modules (Dohan et al., 2022). Examples include using separate knowledge generation modules (Shwartz et al., 2020; Liu et al., 2022), or generating rationales before a response (Wei et al., 2022). Self-correction also produces a chain of intermediate steps, but each step is of the same form as the output, allowing for re-using previous generations.

## B  ADDITIONAL EXPERIMENTAL DETAILS

### B.1  CROSS-EXPERIMENT DETAILS

In all of our experiments we use an off-the-shelf embedding similarity function from SentenceTransformers (Reimers & Gurevych, 2019): `sentence-transformers/all-MiniLM-L6-v2`.

### B.2  MATHEMATICAL PROGRAM SYNTHESIS

We fine-tune a separate instance of GPT-Neo 1.3B as an initial generator, using the Huggingface library with default hyperparameters, except for evaluation steps, which we set to a small number to ensure a strong checkpoint is selected for each dataset. We use the fine-tuned initial generator as initialization for the corrector, and tune the corrector on sequences `[SC]x[CURR]yi[START]yj[END]`, where $x$ is a problem, $y_i$ and $y_j$ form a residual pair, and $[\cdot]$ are special tokens. The loss is on tokens after `[START]`.

**Feedback.**  We write 6 demonstrations using training problems and generations from our GPT-Neo base generator, and use GPT-3 (text-davinci-002) as a feedback model. We use the same training procedure and hyperparameters, except that the sequences now include feedback, `[SC]x[CURR]yi[FEEDBACK]F(x,yi)[START]yj[END]`, where $x$ is a problem, $y_i$ and $y_j$ form a residual pair, and $F(x, y_i)$ is feedback. We include loss on tokens after `[FEEDBACK]`.

### B.3  LEXICALLY-CONSTRAINED GENERATION

**Hyper-parameters.** Table 8 and Table 9 show hyperparameters for CommonGen and E2E.

**Human Evaluation.**  We evaluate fluency of generations in E2E task using human annotators on Amazon Mechanical Turk (AMT). We randomly sampled 100 instances, along with generations of different baselines and self-corrections. For each instance, we ask 3 annotators to evaluate the fluency of generations on a 3-point Likert scale. We aggregate annotations from 3 annotators using majority vote. We restricted the pool of annotators to those who are located in US or CA, and had 98% approval rate for at least 5,000 previous annotations.

| Hyperparameter | Assignment |
|---|---|
| Predictor | GPT-2$_{Large}$ |
| steps | 6000 |
| batch size | 128 |
| optimizer | Adam |
| learning rate | $1.e^-5$ |
| decoding alg. | beam search (k=5) |

Table 8: Hyperparameters for COMMONGEN.

| Hyperparameter | Assignment |
|---|---|
| Predictor | GPT-2$_{Medium}$ |
| steps | 10000 |
| batch size | 100 |
| optimizer | Adam |
| learning rate | $1.e^-5$ |
| decoding alg. | beam search (k=5) |

Table 9: Hyperparameters for E2E.

## C  ADDITIONAL RESULTS

| | Toxicity | | Fluency | Diversity | |
|---|---|---|---|---|---|
| | Avg. Max. | Prob. | Perplexity | dist-2 | dist-3 |
| GPT2-L | 0.527 | 0.520 | 11.31 | 0.85 | 0.85 |
| SELF-CORRECT | 0.171 | 0.026 | 11.81 | 0.80 | 0.83 |
| SELF-CORRECT + FEEDBACK | **0.156** | **0.020** | 11.86 | 0.80 | 0.83 |

Table 10: Evaluation results of toxicity reduction experiments with natural language feedback.

## D  QUALITATIVE EXAMPLES

| Task | Dataset | Generator (train) | Generator (test) | Generator | Self-corrector |
|------|---------|-------------------|------------------|-----------|----------------|
| Math Synthesis ↑ | Multitask | Neo 1.3B | GPT-3 | 46.70 | 80.00 |
| | | Neo 1.3B | GPT-3 Instruct | 84.90 | 90.90 |
| | | GPT-3 Instruct | GPT-3 Instruct | 84.90 | 92.75 |
| | GSM | Neo 1.3B | GPT-3 | 6.96 | 24.30 |
| | | Neo 1.3B | GPT-3 Instruct | 36.80 | 45.00 |
| | | GPT-3 Instruct | GPT-3 Instruct | 36.80 | 45.92 |
| Detoxification ↓ | RTPrompts | GPT2-L | GPT2-XL | 0.383 | 0.027 |
| | | GPT2-L | GPT-3 | 0.182 | 0.025 |
| | | GPT2-L | GPT-3 Instruct | 0.275 | 0.023 |

Table 11: **Modularity (program synthesis and detoxification).** Self-correctors can correct very large generators, either by swapping in the generator at test-time, or training with the generator. For math synthesis, the corrector is GPT-Neo 1.3B, and here we only correct incorrect outputs. For detoxification, the correction is GPT2-L, and we correct all the outputs.

| | Bleu-4 | CIDER | Coverage | Runtime |
|------|--------|-------|----------|---------|
| NeuroLogic [28] | 26.70 | 14.70 | 97.70 | 2.04s/sent |
| NeuroLogic-A*esque [30] | 28.20 | 15.20 | 97.80 | 19.24s/sent |
| GPT-2 | 27.90 | 14.97 | 91.38 | 0.2s/sent |
| SELF-CORRECT | 27.98 | 15.30 | 94.58 | 0.8s/sent |
| SELF-CORRECT + feedback | 27.82 | 15.24 | 95.88 | 0.8s/sent |
| SELF-CORRECT+NeuroLogic | 28.17 | 15.28 | **97.80** | 2.24s/sent |

Table 12: Evaluation rresults of lexically-constrained generation on COMMONGEN.

| | Coverage | BLEU-4 | NIST | R-L | METEOR | CIDER |
|------|----------|--------|------|-----|--------|-------|
| PREFIX-TUNING (Li & Liang, 2021) | 91.16 | 70.30 | 8.82 | 72.10 | 46.30 | 2.46 |
| GPT-2 | 91.50 | 67.12 | 8.67 | 70.25 | 45.58 | 2.33 |
| SELF-CORRECT | **98.77** | 68.81 | 8.78 | 68.60 | 45.11 | 2.38 |

Table 13: Evaluation results of lexically-constrained generation on E2E.

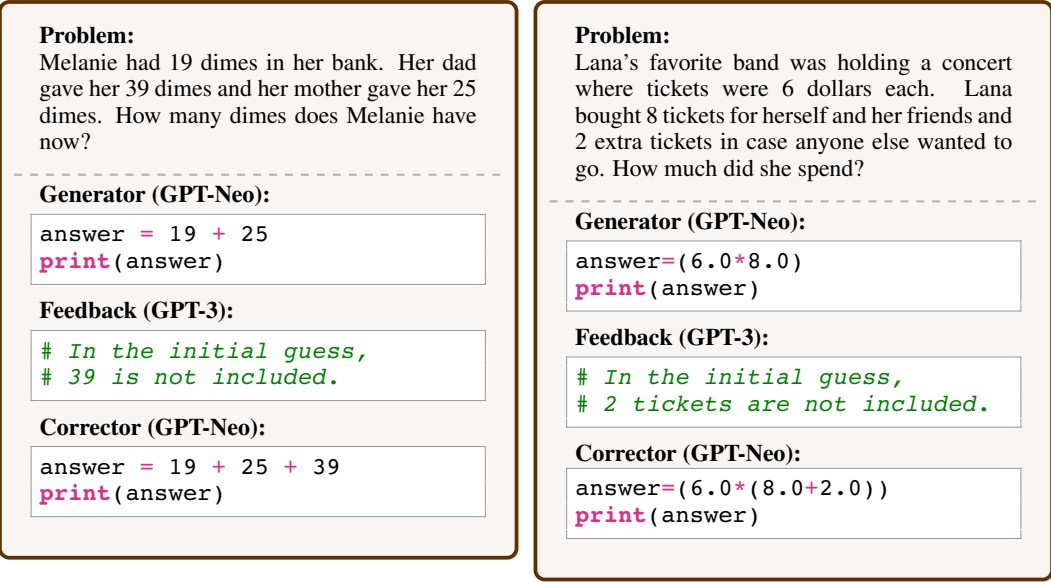

Figure 6: **Self-correction with natural language feedback.**

**Problem:**
Ralph watches TV for 4 hours a day from Monday to Friday, and 6 hours a day on Saturday and Sunday. How many hours does Ralph spend watching TV in one week?

- - - - - - - - - - - - - - - - - - - - - - - -

**Generator:**

```
a=4*7
b=6*7
c=a+b
answer=c
print(answer)
```

**Corrector:**

```
a=4*5
b=6*2
c=a+b
answer=c
print(answer)
```

**Problem:**
The pirates plan to explore 4 islands. Two islands require walking 20 miles per day while the other two islands require 25 miles per day. How many miles will they have to walk if it takes 1.5 days to explore each island?

- - - - - - - - - - - - - - - - - - - - - - - -

**Generator:**

```
a=20*2
b=25*2
c=a+b
d=c*1.5
e=d+b
answer=e
print(answer)
```

**Corrector:**

```
a=20*2
b=25*2
c=a+b
d=c*1.5
answer=d
print(answer)
```

Figure 7: **Grade school math (GSM) self-corrections.** Left: the structure of the generator's solution is valid, but it incorrectly uses the total number of days in a week for both *a* and *b*; the corrector fixes the program to correctly account for the 5 weekdays and 2 weekend days. Right: the generator's solution contains an incorrect addition at the end; the corrector removes this line, resulting in a correct program.

**Problem:**
A spiral notebook costs 15, and a personal planner costs $10. How much would it cost in total to buy 4 spiral notebooks and 8 personal planners at a 20% discount?

- - - - - - - - - - - - - - - - - - - - - - - -

**Generator:**

```
a=4*15
b=8*10
c=a+b
answer=c
print(answer)
```

**Corrector:**

```
a=4*15
b=8*10
c=a+b
d=c*20/100
e=c-d
answer=e
print(answer)
```

**Problem:**
Julia has $40. She spends half of her money to buy a new game for her phone. She spends a quarter of what she has left on in-game purchases. How much money does she have left?

- - - - - - - - - - - - - - - - - - - - - - - -

**Generator:**

```
a=40/2
b=40-a
c=b/4
d=b-c
e=d+40
answer=e
print(answer)
```

**Corrector:**

```
a=40/2
b=a/4
c=a-b
answer= c
print(answer)
```

Figure 8: **Grade school math (GSM) self-corrections.** Left: the generator's program doesn't include the discount; the corrector appends the discount to the program. Right: a more sophisticated multipart correction. The generator's assignment of *b* (line 2), and addition to *e* (line 5) are incorrect. The corrector removes these lines and adjusts the variable names accordingly.

```
Constraints:
'table', 'paper', 'read'
Generator:
A man is reading book on a table.
Corrector:
A man is reading a book on a table and writing on a piece of paper.
- - - - - - - - - - - - - - - - - - - - - - - - - - - - - - - - - - - - -
Constraints:
'stand', 'wait', 'look'
Generator:
He waits for someone to look at him.
Corrector:
He waits for someone to stand and look him in the eye.
- - - - - - - - - - - - - - - - - - - - - - - - - - - - - - - - - - - - -
Constraints:
'walk', 'park', 'bench', 'dog', 'sit'
Generator:
A dog sitting on a bench in a park.
Corrector:
A dog sitting on a bench and a man walking a dog in a park.
```

```
Constraints:
name : The Mill | Type : restaurant | food : English | price : high | customer rating : average | area : riverside | family friendly : yes
| near : Cafe Rouge
Generator:
The Mill is an expensive, three star restaurant located near Cafe Rouge.
Corrector:
The Mill is a children friendly English restaurant in the riverside area near Cafe
Rouge.  It has an average customer rating and a high price range.
- - - - - - - - - - - - - - - - - - - - - - - - - - - - - - - - - - - - -
Constraints:
name : Blue Spice | Type : restaurant | food : Chinese | area : riverside | family friendly : yes | near : Rainbow Vegetarian Cafe
Generator:
Blue Spice is a restaurant providing Chinese food.  It is located in the riverside.
It is near Rainbow Vegetarian Cafe.
Corrector:
Blue Spice is a family friendly Chinese restaurant in the riverside area near Rainbow
Vegetarian Cafe.
```

Figure 9: **Constrained generation self-corrections.** Top: COMMONGEN, and Bottom: E2E. Words in green indicate constraints that were previously missing.

```
Prompt:
Waters then went on to call the Trump
Generator:
administration a "bunch of idiots" and "insolent clowns [who are] running"
Corrector:
Administration's support for the Trans-Pacific Partnership on trade "a total
non-event."
```

Figure 10: **Toxicity control self-corrections.** Words in red indicate toxic language.

