# OpenReview forum: "Generating Sequences by Learning to Self-Correct"
_ICLR.cc/2023/Conference — ICLR 2023 poster_

### Official Review · Reviewer_E4Ei · 2022-10-24

**Confidence:** 4
**Correctness:** 2
**Technical Novelty And Significance:** 3
**Empirical Novelty And Significance:** 3
**Recommendation:** 6

**Clarity, Quality, Novelty And Reproducibility:**

The paper is very clearly written, has high quality, is novel and practical. The
authors will make the code publicly available upon acceptance, making it easier
to reproduce the results.

**Strength And Weaknesses:**

### Strengths

The paper is built upon a good idea with practical applicability: the method allows to
correct large language models using relatively small computational resources.

The evaluations are also comprehensive with different tasks, baselines, and
ablation studies, and the obtained results are excellent.

### Weaknesses

My first concern is about self-correction. I believe that the model corrects its
mistakes and not itself: it does not reprogram itself to work more efficiently.
It is not different in this respect from a genetic algorithm which improves
subsequent generations, especially if we combine it with a neural network. I
don't see what the presented method adds in comparison to warrant naming
it "self-correction". Also, naming the method "self-correction" is too general.

My second concern is about the explanation of the algorithm in 2.1, it could be
made clearer. In the first paragraph, "(e.g., a classifier)" doesn't help to
understand what $v(y)$ is. The second paragraph states that "the algorithm
collects a pool of generations, groups them and selects pairs of generation that
increase in value and are nearby". I think this could be expanded to be more
helpful and precise. The authors could state what the generations are (and that
the first generation is from the outputs of the base generator). The algorithm
doesn't really group the generations: as far as I understand it works on subsets
of the data pool, where each subset corresponds to an input example and contains
(among other things) the outputs of the base generator and the iteratively applied corrector for
that input. Lastly, the algorithm selects pairs of outputs for the same input
which could be from the same generation.

The organization of 2.1. could also be improved somewhat, as the ordering of the
phases does not correspond to the order in which they happen in the algorithm.
Particularly, I would move Exploration after Initialization and before Pairing.

Minor concerns:
- I think it's a stretch that self-corrector paired with NeuroLogic outperforms
  NeuroLogic-A* in Table 2.
- The last line of page 8 should say that Figure 6 is in the Appendix.

**Summary Of The Paper:**

The paper introduces a method to correct the outputs of language models using
another language model in order to satisfy semantic constraints. A function
$v(y)$ is needed to measure the quality of the output hypotheses $y$. The
corrector is trained to improve this quality while staying close to the original
hypothesis. The training is done on value-improving pairs: pairs of outputs $(y,
y')$ from the data pool which belong to the same input and where there is an
improvement in quality ($v(y) < v(y')$). The data pool is initialized by the
base generator but is continously expanded with the corrected outputs, so the
corrector can learn to correct its own output even further in the case of
multiple iterations. It is also possible to provide feedback to the corrector
which improves the results.

The authors study the model and obtain state-of-the-art results on three diverse
tasks with different constraints: a mathematical program synthesis task, a text
generation task given lexical constraints (which words should be present in the
text), and a text generation task where the generated text should not be toxic.



**Summary Of The Review:**

I really liked the paper and I think it could have high impact. I have concerns
about the concept of "self-correction" and about the clarity of the exposition
of the algorithm. If those are addressed I'm going to significantly raise my
score.

-------------------------------------------------------------------------------

Update:

My concerns were paritally addressed, so I'm raising my score.

---

> ### Author Response · Authors · 2022-11-15
> **Author Response**
>
> Thank you for your review and positive comments on the good idea and practical applicability, novelty, and comprehensive evaluation. Please see our comments about naming and exposition.
>
> - Re "My first concern is about self-correction. I believe that the model corrects its mistakes and not itself: it does not reprogram itself to work more efficiently. It is not different in this respect from a genetic algorithm which improves subsequent generations, especially if we combine it with a neural network. I don't see what the presented method adds in comparison to warrant naming it "self-correction". Also, naming the method "self-correction" is too general."
>
> The name Self-Correction captures the core idea behind our method, as shown in Equation 1 and Figure 1. Self-Correction is a framework which includes a generator module and a corrector module, where the corrector learns to correct the mistakes. Hence, "self" refers to the compound framework. Intuitively, we can interpret the two-part system as improving itself by learning from its own mistakes. We’re also open to changing to an alternative name if it is a significant blocker to having the paper accepted.
>
> - Re "My second concern is about the explanation of the algorithm in 2.1, it could be made clearer. In the first paragraph, "(e.g., a classifier)" doesn't help to understand what v(y) is."
>
> Thank you, we updated the explanation of v(y).
>
> - Re: "The second paragraph states that "the algorithm collects a pool of generations, groups them and selects pairs of generation that increase in value and are nearby". I think this could be expanded to be more helpful and precise. The authors could state what the generations are (and that the first generation is from the outputs of the base generator)."
>
> To clarify we changed the word “groups” to “pairs” in the second paragraph so that it follows Figure 2. We believe that Figure 2 shows the high level concept, then for readers who are interested in further details, each step is written precisely in the subsequent paragraphs and equations (equations 2-5). That said, we’re glad to make any other changes needed to make it clearer.
>
> - Re: "The algorithm doesn't really group the generations: as far as I understand it works on subsets of the data pool, where each subset corresponds to an input example and contains (among other things) the outputs of the base generator and the iteratively applied corrector for that input. Lastly, the algorithm selects pairs of outputs for the same input which could be from the same generation."
>
> In practice we group the pairs based on x. We iterate through the x’s (in a random order) and sample a pair for each x according to equation 4. We updated the presentation to say $x\sim \mathcal{U}(X)$ prior to equation 4. Thanks for catching that point!
>
> - Re: "The organization of 2.1. could also be improved somewhat, as the ordering of the phases does not correspond to the order in which they happen in the algorithm. Particularly, I would move Exploration after Initialization and before Pairing."
>
> To clarify, in practice, after initialization we do a round of pairing and learning prior to exploration. Then we iterate between exploration, pairing, and learning. We updated the Algorithm 1 to reflect this ordering, instead of the paragraph ordering. Thanks for pointing that out!

---

> > ### Comment · Reviewer_E4Ei · 2022-12-02
> > **Thank you for your response and corrections**
> >
> > I believe that the paper got easier to understand and better in general.
> >
> > I'm still not sure about calling the method Self-correction. What alternative would you suggest?

---

> > > ### Author Response · Authors · 2022-12-12
> > > **Author reply**
> > >
> > > Thank you! Regarding the name, we would suggest "Prediction-Correction". Although we do think Self-Correction captures the ideas presented in the paper well, Prediction-Correction may make the distinction between the generator (i.e. predictor) and corrector more apparent.
> > >
> > > We are glad that the paper is now easier to understand and better in general. If you think our comments and updates have addressed your concerns, could you please consider raising your rating of the paper? Thank you!

---

### Official Review · Reviewer_Udn3 · 2022-10-24

**Confidence:** 5
**Correctness:** 3
**Technical Novelty And Significance:** 2
**Empirical Novelty And Significance:** 2
**Recommendation:** 6

**Clarity, Quality, Novelty And Reproducibility:**

The writing is generally clear, except that the similar function design is unclear.

The presented self-correction approach is sound and empirically effective. However, it is not a novel approach and has been well-studied in neural program synthesis domain, and the authors did not properly discuss the related work.

The authors promise to release the source code upon paper acceptance.

**Strength And Weaknesses:**

Strengths:

1. Iteratively decoding is a promising direction to improve the solution quality for a wide range of tasks, including sequence generation.

2. The evaluation covers different domains and shows good empirical results.

Weaknesses:

1. Learning input correction is not a novel approach. This work completely ignores a long line of research on learning to repair in the code domain. For example, there are existing works that learn a neural debugger to repair the prediction of a base program synthesizer [e.g., 1, 2, 3, 4], and propose neural networks for stand-alone program repair tasks (among others, [5, 6] are closely related in terms of the approach design). The authors should provide a proper discussion of related works in this space.

2. The evaluation setup is not convincing. The base generators utilize a small beam size, and sometimes with greedy decoding. A fair comparison is to increase the number of samples from the base generator, and see whether the corrector improves the performance with the same number of samples in total.

3. Also, it is unclear whether the approach improves over the SOTA results. For example, on GSM dataset and some other math benchmarks, the SOTA approach is self-consistency [7], where the best result on GSM is 78%. The authors should evaluate their self-correction method upon better base generators and compare the performance.

4. In Equation 4, it is unclear how the similarity function is defined.

[1] Gupta et al., Synthesize, Execute and Debug: Learning to Repair for Neural Program Synthesis, NeurIPS 2020.
[2] Balog et al., Neural Program Synthesis with a Differentiable Fixer.
[3] Fu et al., Coda: An End-to-End Neural Program Decompiler, NeurIPS 2019.
[4] Le, Wang et al., CodeRL: Mastering Code Generation through Pretrained Models and Deep Reinforcement Learning, NeurIPS 2022.
[5] Yasunaga and Liang, Graph-based, Self-Supervised Program Repair from Diagnostic Feedback, ICML 2020.
[6] Yasunaga and Liang,  Break-It-Fix-It: Unsupervised Learning for Program Repair, ICML 2021.
[7] Wang et al., Self-consistency improves chain of thought reasoning in language models.

**Summary Of The Paper:**

This paper presents self-correction approach for sequence generation. Specifically, given a sequence decoded by the base generator, they train a corrector to generate another sequence, with the goal of achieving a better score than the input sequence. They design the self-corrective learning algorithm to train the corrector, where they select sequence pairs for training that: (1) the target sequence improves the score; and (2) the target sequence stays relatively similar to the input sequence. They also evaluate a setting where the corrector can leverage additional natural language feedback for correction. They evaluate their approach on 3 tasks: mathematical program synthesis, lexically-constrained generation, and toxicity control. The results demonstrate that adding a corrector improves the results over the base generator.

**Summary Of The Review:**

Iteratively decoding is a promising direction. However, this approach is not novel and has been well-studied in neural program synthesis domain, and the authors did not properly discuss the related work. Meanwhile, the evaluation setting is not convincing and misses some important comparisons and ablation studies. Therefore, I recommend rejecting this submission.


--------
I thank the authors for their response, and I updated my score.

---

> ### Author Response · Authors · 2022-11-15
> **Author Response**
>
> Thank you for your detailed review and comments on the promising direction, sound and empirically effective approach. Please see our comments regarding novelty and evaluation, and our updated discussion of the related work.
>
> - Re: "Learning input correction is not a novel approach. This work completely ignores a long line of research on learning to repair in the code domain. For example, there are existing works that learn a neural debugger to repair the prediction of a base program synthesizer [e.g., 1, 2, 3, 4], and propose neural networks for stand-alone program repair tasks (among others, [5, 6] are closely related in terms of the approach design). The authors should provide a proper discussion of related works in this space."
>
>
> Thank you for pointing out work in the code domain that we overlooked. Indeed, these works warrant a discussion and references; we have added references early on in the introduction, a discussion in the related work, and a detailed discussion in a new extended related work section in the Appendix.
>
> Regarding novelty, at a high level the Self-Correction framework provides a new formalism and algorithm that unifies several ideas (learning from scalar feedback, explicit feedback, learning a corrector), and generalizes to diverse generation tasks. More specifically, our novelty lies both in the method design as well as the problem setting.
>
> Methodologically, the closest work in the code domain is [6]: Break-It-Fix-It.
> Our work differs from [6] in two key respects:
>
> 1.  **Data assumptions**: Unlike [6], self-correction does not assume a mechanism for generating synthetic negatives, a dataset of negatives, or a backwards model that generates negatives. We demonstrate that self-correction can be effective without these components, which is important because engineering these for each new task can be prohibitive.
> 2.  **New mechanisms**: forming value-improving pairs from model generations (Equation 3), proportional sampling (Equation 4), and exploration were all found to be important designs for training the corrector (Tables 6,7). Integrating these ideas into a single algorithm that works in diverse generation tasks is novel and important for being able to build off of in the future.
>
> Second, regarding novelty in the problem setting, self-correction generalizes to a diverse set of generation tasks. The setting covers three very different types of problems: improving a fine-tuned generator with a 0/1 value on math problems, open-ended generation with a continuous value, and constrained generation with a scalar value. Moreover, we explore a new setting of correcting the outputs of a very large language model. Both of these are new from a correction perspective.
>
> We have added a discussion into the extended related work in the Appendix. We acknowledge that there was missing discussion in the original version, and we have updated it accordingly.
>
> - Re "The base generators utilize a small beam size, and sometimes with greedy decoding. A fair comparison is to increase the number of samples from the base generator, and see whether the corrector improves the performance with the same number of samples in total."
>
> We use the decoding procedures found in prior work for each task, and at test time all of the baselines use the same decoding procedures for fair comparison.
>
> In toxicity, the metrics account for multiple samples: they draw 25 samples and measure the average maximum toxicity and the empirical toxicity probability. For lexically constrained generation, we use beam search with the same settings as prior work.
>
> For math, we use greedy decoding as in [Mishra et al 2022]. However, we ran an additional experiment to address your concern. Using the baseline GPT-NEO 1.3B model, we sample 10 outputs plus the greedy output and verify all 11 outputs. In contrast, self-correct* only uses 2 verifications. Self-correction outperforms the baseline even with 2/11=18% of the verification calls:
>
> | Dataset      | Method | Correct|
> | ----------- | ----------- |----------- |
> Multitask|Baseline (best of 10 + greedy) |64.90|
> Multitask|Self-Correct* (greedy + greedy correction)|**78.24**|
>
> - Re: "Also, it is unclear whether the approach improves over the SOTA results. For example, on GSM dataset and some other math benchmarks, the SOTA approach is self-consistency [7], where the best result on GSM is 78%. The authors should evaluate their self-correction method upon better base generators and compare the performance."
>
> Self-Correction outperformed an extensive list of state-of-the-art methods on toxicity, and outperformed the state-of-the-art on math program synthesis for publicly available models in the range of 1-3B.
>
> Improving over SOTA was not the purpose of the modularity experiments; instead, these experiments demonstrated a practical use case: getting an output from a large model with a standard decoding procedure, then correcting it with a much smaller model.

---

> > ### Author Response · Authors · 2022-11-15
> > **Author Response (continued)**
> >
> > - Re "In Equation 4, it is unclear how the similarity function is defined."
> >
> > Thank you for pointing that out! We added a comment to the Appendix. We use the same embedding-based similarity function in all of our experiments ('sentence-transformers/all-MiniLM-L6-v2'), though users are free to experiment with others as needed.
> >
> > To conclude, we thank the reviewer for their extensive and fine-grained review. Your comments on the related work help to clarify our contribution and to connect it with prior ideas in the code domain.

---

### Official Review · Reviewer_iywo · 2022-10-26

**Confidence:** 4
**Correctness:** 3
**Technical Novelty And Significance:** 4
**Empirical Novelty And Significance:** 4
**Recommendation:** 8

**Clarity, Quality, Novelty And Reproducibility:**

I consider the idea how to generate data for training self-correcting models the main novelty. This is a significant and clearly expressed idea that might help a wide range of applications.

The paper is written very well and the authors promise to release the source code upon acceptance of the paper.

The authors might find that this paper is related: https://arxiv.org/pdf/2003.10555.pdf.

**Strength And Weaknesses:**

Pros: Great idea, generally great set of experiments.

Cons: I was not able to understand the baselines. Since this paper assumes a dataset with ground truth answers. So, an obvious baseline for this approach would be a model that is fine-tuned on the ground truth. I was not able to understand which of the baselines were fine-tuned and which of them relied on few-shot prompting. I kindly ask the authors to clarify this point (or provide the additional baseline). I will raise my score significantly, if this can be clarified.

**Summary Of The Paper:**

This paper suggests a new method to generate sequences with language models. Instead of sampling directly from the model, the process first generates a candidate and then revises the candidate using a "self-correction" model (possibly in multiple rounds). The core of the paper is an elegant method to generate training data for self-correcting models. Starting with a generative language model, the method generates candidate answers. The idea is to select the candidate answer that is "wrong" (i.e. of low quality) but most similar to the correct answer (=ground truth, which is assumed to be available for the training set). This forms a new training example that consists of the original input, the flawed candidate and the improved candidate (=ground truth).

The similarity between the selected candidate answer and the correct answer is important to make the correction related to the originally generated sequence, which the paper confirms in an ablation.

**Summary Of The Review:**

Probably a great paper, but with a serious question around the baselines. I will increase my score if this concern can be addressed.

---

> ### Author Response · Authors · 2022-11-15
> **Author Response**
>
> Thank you for your detailed review and comments on the novelty of our training methodology and quality of our experiments. Please see our clarification regarding the baselines below.
>
> We ensure strong baselines for all of our settings; to clarify:
>
> 1. In the main math experiments (Table 1), each baseline is fine-tuned on the ground truth dataset. For example, GPT-NEO 1.3B refers to fine-tuning GPT-NEO on the dataset in math. The other baselines are also fine-tuned. We added a clarification in our updated version.
> 2. In lexically-constrained generation (Table 2), each baseline is fine-tuned on the ground-truth dataset. The neurologic baselines use an additional decoding algorithm on top of the fine-tuned GPT-2 baseline.
> 3. In toxicity (Table 3), there is no ground-truth dataset of non-toxic continuations. We follow previous works (Liu et al. 2021) and report the toxicity of off-the-shelf GPT-2 as well as an extensive list of prior methods.
> 4. In the modularity experiments, we use GPT-3 few-shot to reflect how GPT-3 is used in practice. We then train a corrector using GPT-Neo (math) or GPT2 (toxicity).

---

> > ### Comment · Reviewer_iywo · 2022-12-04
> > **Updated score**
> >
> > Thanks for your response. This addresses my concern and I raised the score.

---

> > > ### Author Response · Authors · 2022-12-12
> > > **Thank you**
> > >
> > > We are glad that the concerns were addressed, and thank you for updating your score!

---

### Official Review · Reviewer_qE6M · 2022-10-27

**Confidence:** 3
**Correctness:** 3
**Technical Novelty And Significance:** 3
**Empirical Novelty And Significance:** 3
**Recommendation:** 5

**Clarity, Quality, Novelty And Reproducibility:**

The authors should further clarify the method design and the experimental settings. The overall quality of this paper is OK. But in my view, the novelty of the proposed method is somewhat limited from the perspective of correctors. The reproducibility of this paper is degraded due to the lack of codes.

**Strength And Weaknesses:**

Strengths:

1) This paper is well organized and easy to follow.
2) The proposed method can be applied to a wide range of text generation tasks. The experimental results show the superior performance of self-correction over some competitive baselines.

Weaknesses:

1) The name of the method “self-correction” is confusing for me, because the authors train a separate corrector to improve the base generator. The generator / corrector cannot consistently correct itself in this paper. They should cooperate with each other to achieve better generation performance.

2) From the perspective of correctors, the proposed method seems to train a text editing model (corrector) via pseudo-labeled data generated by a pre-trained model (generator). Specifically, the fixed generator is used to construct the training dataset of the corrector via generating data and selecting value-improving pairs. Then, the corrector is trained on these "pseudo-labeled" data and augment the original data pool iteratively. Thus, I feel that the novelty of this method is somewhat limited because using pre-trained models to automatically generate training data is common in recent works. I don’t find any specific design when training the corrector.

3) The feedback has been mentioned for many times in this paper. But this part is individual compared with the whole design. I don’t find any specific module to properly incorporate the feedback signal into the corrector.

4) The experimental setting may be unfair because the corrector has a relatively large amount of model parameters. Thus, the total number of parameters in self-correction (including the generator and the corrector) is significantly larger than that of other baselines.

**Summary Of The Paper:**

This paper proposes a self-correction method which trains a corrector to iteratively correct imperfect generation results. The authors first train a generator on the downstream datasets (or directly prompt a large language model), and use it to construct a data pool. Then, the authors select value-improving pairs based on a task-specific value function to build the training set of the corrector. Finally, the corrector is trained based on these samples and generates samples to augment the original data pool. Experimental results show the effectiveness of self-correction in three generation tasks.

**Summary Of The Review:**

The proposed method can adapt to various text generation tasks and achieve good performance. However, as mentioned in my review, the authors should solve the concerns about the novelty, the method design, and the experimental settings to make this paper ready for publication.

---

> ### Author Response · Authors · 2022-11-15
> **Author Response**
>
> Thank you for your review and positive comments on the wide applicability and strong performance of our method over competitive baselines. Please see our responses related to the naming, novelty, and experimental settings below.
>
> - Re: "The name of the method “self-correction” is confusing for me, because the authors train a separate corrector to improve the base generator. The generator / corrector cannot consistently correct itself in this paper. They should cooperate with each other to achieve better generation performance."
>
> The name Self-Correction captures the core idea behind our method, as shown in Equation 1 and Figure 1. Self-Correction is a framework which includes a generator module and a corrector module where the corrector learns to correct the mistakes. Hence, "self" refers to the compound framework. We agree with the reviewer that neither module solves the problem on its own, they cooperate as a system to solve the problem together. We show that this form of cooperation achieves better generation performance empirically. Meanwhile, we’re open to changing to an alternative title if it is a significant blocker to having the paper accepted.
>
> - Re: novelty, "Thus, I feel that the novelty of this method is somewhat limited because using pre-trained models to automatically generate training data is common in recent works. I don’t find any specific design when training the corrector.":
>
> Regarding novelty, at a high level, the Self-Correction framework provides a new formalism and algorithm that unifies several ideas (learning from scalar feedback, explicit feedback, learning a corrector), and generalizes to diverse generation tasks. More specifically, our novelty lies both in the method design as well as the problem setting.
>
> First, regarding the novelty in method design, we propose a new way to train a corrector for our general problem setting. In particular, forming value-improving pairs from model generations (Equation 3), proportional sampling (Equation 4), and exploration were all found to be important designs for training the corrector (Tables 6,7). Integrating these ideas into a single algorithm that works in diverse generation tasks is novel and important for being able to build off of in the future. Our main idea is orthogonal to lines of work on data augmentation. Data generation is merely part of our method design to train the corrector.
>
> Moreover, regarding the novelty in problem setting, self-correction generalizes to a diverse set of generation tasks, versus previous work which focused on a specific correction setting. Our setting covers three very different types of problems: improving a fine-tuned generator with a 0/1 value on math problems, open-ended generation with a continuous value, and constrained generation with a scalar value. Moreover, we explore a new setting of correcting the outputs of a very large language model. Both of these are new from a correction perspective.
>
> For additional, lower-level details on how these differ from previous work, please see our response to R3. In the Appendix we have updated the related work to clarify our contributions, and we will integrate these comments into the main text for the final revision.
>
> - Re: "The feedback has been mentioned for many times in this paper. But this part is individual compared with the whole design. I don’t find any specific module to properly incorporate the feedback signal into the corrector."
>
> Using feedback is a critical component of self-correction. Self correction takes two types of feedback: 1) scalar feedback from a classifier, which is used to form value-improving pairs for learning, which enables the corrector to generate the value-improving corrections at test time 2) natural language feedback, which is fed as an additional input to the corrector to provide a richer signal for how to correct.
>
> The ability to integrate scalar or natural language feedback demonstrates the flexibility of our self-correction framework. In contrast with other learning frameworks that only use scalar feedback as a reward (for example, RL as it is typically used in language generation), self-correction can incorporate additional natural language feedback, which can provide an additional learning signal. Indeed, this led to performance improvements in our experiments. For example, in toxicity, we provide additional natural language feedback to the corrector (e.g., "decrease toxicity in profanity") . As evidenced by the improved metrics (see Table 5), it learns to use the natural language feedback to make generations less toxic compared to without the feedback.

---

> > ### Author Response · Authors · 2022-11-15
> > **Author Response (continued)**
> >
> > - Re "The experimental setting may be unfair because the corrector has a relatively large amount of model parameters. Thus, the total number of parameters in self-correction (including the generator and the corrector) is significantly larger than that of other baselines."
> >
> > Self-correction outperformed baselines even after accounting for the total parameter count (i.e. generator+corrector). In math, the self-corrector outperforms a 2.7B parameter model, which is larger than the generator plus corrector (2.6B). In toxicity reduction, the other strong baselines use at least as many parameters— PPO and Quark use 2 copies of the model (original policy, current policy) for their KL-divergence constraint, and PPO uses an additional third copy for its value function. We added these comments in the paper.

---

### Author Response · Authors · 2022-11-15
**Author Summary Comment**

We thank the reviewers for their thoughtful and helpful comments. The reviewers found Self-Correction to be a great idea with novelty (R2, R4), promising direction (R3), applicable to a wide range of tasks (R1,R2,R3), with a great set of experiments (R2), comprehensive evaluation (R2,R4) and good performance (R1,R3,R4). Moreover, the new setting of correcting large language models using small amounts of resources was also noted as holding practical applicability (R4).

We’d like to highlight that among other contributions, Self-Correction provides a new formalism and algorithm that unifies several ideas (learning from scalar feedback, explicit feedback, learning a corrector), and generalizes to new, diverse generation tasks.

We have updated the submission and addressed the feedback from each reviewer below, including the adjustments that we made in the revision.

---

### Author Response · Authors · 2022-12-12
**Awaiting replies from two reviewers**

Dear Area Chairs,

Since the rebuttal period began, we have posted replies to address the concerns of all reviewers, added a new extended related work section, conducted an additional experiment, and integrated the reviewers' feedback into an updated revision. However, we have not heard any further feedback from the two reviewers whose questions and concerns we spent the most time addressing (qE6M and Udn3).

We hope that during the discussion period, you could further encourage them to respond to our replies and perhaps reconsider their ratings of the paper.

Thank you,
Authors

---

### Decision · Program_Chairs · 2023-01-20

**Decision:**

Accept: poster

**Justification For Why Not Higher Score:**

A couple issues are worth addressing a bit more for this work to get a higher score, in particular on related work and the setup of the experiments, regarding the base + corrector setup potentially using more model parameters and more evaluation budget.

**Justification For Why Not Lower Score:**

Overall the idea proposed in this paper is valuable, and experiment results back that up (though with some room for improvement).

**Metareview: Summary, Strengths And Weaknesses:**

This paper presents the idea of training a corrector for sequence generation, which can be used to improve samples from a base model.  The authors designed a simple way of creating training data for the corrector by sampling from the base model and creating pairs in the direction of improvement.  Extensive experiments are done for a variety of domains that show the promise of this approach.

Overall the reviewers and myself liked the idea, but there are some questions that are worth addressing a bit more.  One is related work - following the suggestions by the reviewers the authors can consider putting the work better into the context, in particular given the extensive literature on the topic of editing programs in the program generation / synthesis literature.  The other is the setup for comparisons, the reviewers raised a few important points, that the base + corrector setup (1) uses an additional model and therefore the total parameter count can be significantly higher, and (2) potentially uses more evaluations, which creates an unfair advantage.  The authors acknowledged these in the comments to the reviewers and have made some changes to the paper, but in general these two points should be made very clear and followed consistently in the comparisons.

I'm supporting an acceptance given the proposed idea's simplicity and good empirical results, but do urge the authors to improve the paper and address the issues mentioned above.

**Note From Pc:**

if the above contains the word "oral" or "spotlight" please see: "oral" presentation means -> notable-top-5% and "spotlight" means -> notable-top-25%. As stated in our emails, we are disassociating presentation type from AC recommendations